# Sentinel Lymph Node Biopsy in Breast Cancer Using Different Types of Tracers According to Molecular Subtypes and Breast Density—A Randomized Clinical Study

**DOI:** 10.3390/diagnostics14212439

**Published:** 2024-10-31

**Authors:** Ionut Flaviu Faur, Amadeus Dobrescu, Ioana Adelina Clim, Paul Pasca, Catalin Prodan-Barbulescu, Cristi Tarta, Carmen Neamtu, Alexandru Isaic, Dan Brebu, Vlad Braicu, Catalin Vladut Ionut Feier, Ciprian Duta, Bogdan Totolici

**Affiliations:** 1II^nd^ Surgery Clinic, Timisoara Emergency County Hospital, 300723 Timisoara, Romania; dobrescu.amadeus@umft.ro (A.D.);paul.pasca@umft.ro (P.P.); catalin.prodan-barbulescu@umft.ro (C.P.-B.); tarta.cristi@umft.ro (C.T.); isaicus@gmail.com (A.I.); brebu.dan@umft.ro (D.B.); braicu.vlad@umft.ro (V.B.); duta.ciprian@umft.ro (C.D.); 2X Department of General Surgery, “Victor Babes” University of Medicine and Pharmacy Timisoara, 300041 Timisoara, Romania; catalin.feier@umft.ro; 3Multidisciplinary Doctoral School “Vasile Goldiș”, Western University of Arad, 310025 Arad, Romania; 4II^nd^ Obstetrics and Gynecology Clinic “Dominic Stanca”, 400124 Cluj-Napoca, Romania; clim.adelina@yahoo.com; 5Department I, Discipline of Anatomy and Embriology, “Victor Babes” University of Medicine and Pharmacy, 300041 Timisoara, Romania; 6Doctoral School, “Victor Babes” University of Medicine and Pharmacy Timisoara, Eftimie Murgu Square 2, 300041 Timisoara, Romania; 7Faculty of Dentistry, “Vasile Goldiș” Western University of Arad, 310025 Arad, Romania; neamtu.carmen@uvvg.ro; 8I^st^ Clinic of General Surgery, Arad County Emergency Clinical Hospital, 310158 Arad, Romania; totolici.bogdan@uvvg.ro; 9First Surgery Clinic, “Pius Brinzeu” Clinical Emergency Hospital, 300723 Timisoara, Romania; 10Department of General Surgery, Faculty of Medicine, “Vasile Goldiș” Western University of Arad, 310025 Arad, Romania

**Keywords:** sentinel lymph node, ICG, methylene blue, breast density, Tabar-Gram classification, molecular subtypes, axillary lymph node dissection

## Abstract

**Background:** Sentinel lymph node biopsy (SLNB) has become a method more and more frequently used in loco-regional breast cancer in the initial stages. Starting from the first report on the technical feasibility of the sentinel node method in breast cancer, published by Krag (1993) and Giuliano (1994), the method underwent numerous improvements and was also largely used worldwide. **Methods:** This article is a prospective study that took place at the “SJUPBT Surgery Clinic Timisoara” over a period of 1 year between July 2023 and July 2024, during which 137 underwent sentinel lymph node biopsy (SLNB) based on the current guidelines. For the identification of sentinel lymph nodes, we used various methods, including single traces and also a dual tracer and triple tracer. **Results:** Breast density represents a predictive biomarker for the identification rate of a sentinel node, being directly correlated with BMI (above 30 kg/m^2^) and with an age of above 50 years. The classification of the patients according to breast density represents an important criterion given that an adipose breast density (Tabar-Gram I-II) represents a lower IR of SLN compared with a density of the fibro-nodular type (Tabar-Gram III-V). We did not obtain any statistically significant data for the linear correlations between IR and the molecular profile, whether referring to the luminal subtypes (Luminal A and Luminal B) or to the non-luminal ones (HER2+ and TNBC), with *p* > 0.05, 0.201 [0.88, 0.167]; z = 1.82.

## 1. Introduction

Breast cancer represents a worldwide health issue, being the most common neoplasia among women [1]. Therefore, the early application of various optimal therapeutic methods that have the aim of reducing morbidity and post-therapeutic morbidity represents an important phase. The development of medical technology and the flexibility of surgical interventions depending on molecular parameters led to a significant reduction in surgical invasions, which in turn led to a surgical decrease that caused the Halstedian era to be forgotten [2,3].

The ACOSOG Z0011 trial had a major impact on reducing axillary surgery for patients with T1-2 breast cancer and 1–2 positive sentinel nodes. Sentinel lymph node biopsy (SLNB) has increasingly become a common approach in early-stage breast cancer treatment. Since the initial demonstrations of the sentinel node technique in breast cancer by Krag in 1993 and Giuliano in 1994, the method has undergone significant improvements and is now widely used around the world [4]. By 2030, there is expected to be a clear consensus on whether axillary lymph node dissection (ALND) should be routinely performed alongside SLNB. Meanwhile, the TAXIS, ADARNAT, and ALLIANCE trials are examining the benefits of these procedures in terms of disease-free survival (DFS) and overall survival (OS) for specific patient groups. Currently, the most optimally considered method for performing SLNB is the dual-tracer technique, preferably a colorimetric one (methylene blue—MB) and the radioisotope tracer technique (Technetium-99m) with optimal detection rates. The false negative rate (FNR) in the specialized literature varies between 2% and 28% depending on the tracer used. According to the specialized literature, the detection rate represents percentage values above 85%, regardless of the tracer used [5].

However, there are other indicators that need to be taken into consideration when it comes to the detection rate of the sentinel node, such as the following: body mass index (BMI), tumor dimensions, tumor location, molecular subtyping, and node density. As for breast density, nowadays, there are numerous classifications used to indicate it, with the most common one being the Tabar-Gram, which classifies breast density into five different categories: I—scalloped contours and Cooper’s ligaments, II—evenly scattered terminal ductal lobular units, III—oval-shaped lucent areas, IV—extensive nodular and linear densities, and V—homogeneous structureless fibrosis with convex contours [6,7]. For breast density classification methods, well-known breast imaging and reporting data system (BI-RADS) standards have been used to reduce complexity in breast imaging evaluation and to aid in outcome monitoring. BI-RADS classification categorizes breast density into four classes: fatty, scattered fibroglandular, heterogeneously dense, and extremely dense [8].

### SLNB Recommendations

The histopathological status of the axillary lymphatic system and locoregional represents an important criterion in therapeutic guidance. Neoadjuvant chemotherapy (NACT) has been proven to successfully downstage up to 40% of pre-chemotherapy-documented axillary lymph node (ALN) metastasis and even eradicate biopsy-proven ALN metastasis in 32% of breast cancer (BC) cases. Performing an SLNB for those patients who underwent NACT with cN+ pre-NACT and ypN0 post-Nact is considered today to be the standardized version [9].

This study has the goal of analyzing the detection rate of the sentinel node (SLNB) by using different tracers either in a single version or combined ones (methylene blue—MB, indocyanine green—ICG, MB + ICG, RI + MB, RI + ICG, and MB + ICG + RI) in conjunction with the tumoral molecular subtype, body mass index (BMI), and node density, highlighting of some potential predictive biomarkers with regard to the identification rate of the sentinel node.

## 2. Materials and Methods

This article is a prospective study that took place at the “SJUPBT Surgery Clinique Timisoara” during a period of 1 year between July 2023 and July 2024, with 233 patients registered with a histopathologic and immunohistochemical diagnosis confirmed by breast cancer, out of which 137 patients underwent sentinel lymph node biopsy (SLNB) based on the current guidelines. For the identification of sentinel lymph nodes, we used various methods, including single traces and also a dual tracer and triple tracer.

The patients included in this study were chosen based on the following criteria: above 18 years of age; cT 1-2 N0; ypT1-2N0; complete imagistic–molecular staging; and signed a consent form to participate in this study. On the other hand, the exclusion criteria were as follows: under 18 years of age; cT3-4; ypT1-2 N+; incomplete imagistic–molecular staging; and refusal to participate in this study.

### 2.1. Density Identification

Breast density identification was peculiar for each patient based on the Tabar-Gram mammographic classification. This classification is based on anatomic–mammographic correlations using a 3D technique and has five different patterns depending on the report between the nodular structure in itself and the perinodular adipose tissue. Pattern I includes (a) scalloped contours and Cooper’s ligaments; (b) scattered terminal ductal lobular units (TDLUs) with 1–2 mm nodular densities visible on mammograms; and (c) oval-shaped lucent areas indicative of fatty replacement. Pattern II signifies complete fatty replacement. Pattern III features a prominent duct pattern in the retroareolar region due to periductal elastosis combined with fatty involution. Pattern IV is characterized by extensive nodular and linear densities throughout the breast. Pattern V consists of homogeneous, ground glass-like, structureless fibrosis with convex contour [10].

### 2.2. Approval from the Ethics Commission

All processes used in this study that included human subjects benefited from the approval of the ethics commission according to national and international standards in direct relation to the Helsinki Declaration of 1964. This article does not include studies on laboratory animals. The consent mentioned above was received from, and approved by, each participant in this study. (The Ethics Commission for Research and Development Activities and for Quality Assurance of Clinical Trials of the “Pius Brinzeu’’ Clinical Hospital Timisoara, Nr. 462/18.04.2024).

### 2.3. SLNB Methods

Based on the location of the sentinel node, we opted for single-tracer methods (MB, ICG), dual-tracer methods (ICG + MB/RI + MB/RI + ICG), and also triple-tracer methods (RI + MB + ICG). The general batch was divided into study groups depending on the sentinel node identification method. Therefore, the numbers were as follows: 24 patients with ICG, 23 patients with MB, 18 patients with ICG + MB, 21 patients with RI + MB, 28 patients with RI + ICG, and 23 patients with RI + ICG + MB.

### 2.4. MB (Methylene Blue)

Methylene blue was administered at a concentration of 1% with a 2 mL periareolar injection for both left and right breasts, followed by the Giuliano–Krag method for 5 min. The incision was 2–3 cm, and it was performed at the caudal border of the axilla approximately 10–12 min after the injections.

### 2.5. ICG (Indocyanine Green)

ICG (Verdye 5 mg/mL) was administered at a concentration of 2.5 mg/mL (Kitai technique) with a periareolar injection of 1 mL (2.5 mg) at the level of the Sappey plexus, both for the left and right breasts. This was followed, as well, by the Giuliano–Krag method for a period of 5 min. A Photodynamic Eye Camera was also used to identify the lymphatic trajectory used to locate the SLN and, at the same time, to guide the incision.

### 2.6. ICG + MB (Indocyanine Green + Methylene Blue)

In terms of combined techniques, the dual tracer (ICG + MB) follows the quantity and concentration of the single tracer, conserving the injection in the same situs. For our study, we initially opted for MB injection followed by ICG.

### 2.7. RI + MB (Radioisotope—Tc-99m + Methylene Blue)

Tc-99m, a radioactive colloid, microfiltered 0.22 µm, was injected in fractional doses in a peritumoral way at least 2 h before the surgery. The lymphatic mapping was performed via lymphoscintigraphy as a regular investigation in order to permit the mapping of the surgical incision. SLN identification was performed by using the EuroProbe2. MB injection was performed in the standard way.

### 2.8. RI + ICG (Radioisotope + Indocyanine Green)

The injection of the microfiltered colloid Tc-99m was performed in a conventional way, followed by lymphoscintigraphy. ICC injection was performed using the Kitai technique.

### 2.9. RI + MB-ICG (Triple Tracer)

This triple tracer method represents the conventional injection method of the 3 tracers via the techniques presented above.

### 2.10. Histopathologic Examination of SLNB

The examination of the excision tools was performed in a standardized way by an anatomical pathology specialist who specializes in breast cancer tumors according to the American Joint Committee on Cancer and ASCO guidelines. The identification of the numbers and dimensions of the nodes were regular parameters, which is why the excision tools were sectioned at 2 mm and reported to the longitudinal axis.

The clustering ITC-isolated tumor cells were defined as single cells or small clusters of cells not >0.2 mm and no more than 200 cells in a single cross-section. Micrometastasis was defined as the presence of tumoral cells at the SLNB level with dimensions between 0.2 and 2 mm. Macrometastasis was defined as the presence of tumoral cells at the SLNB level with dimensions higher than 2 mm.

### 2.11. Immunohistochemistry Analysis and Molecular Typing

A molecular stratification was performed according to the St. Gallen classification, and we also analyzed the hormonal status of the patients (ER-estrogen receptor and PR-progesterone receptor) and the HER2 status and Ki 67 index. We considered the hormonal expression of both ER and PR positive if a percentage of over 1% was obtained. If HER2 was evaluated as 3+ by immunohistochemical staining or recorded over 2.0-fold growth by fluorescence in situ hybridization, the HER2 expression was considered positive. The Ki67 value was defined as the proportion of positively stained cells (500–1000) among the total number of cancer cells in the invasive front of the tumor [11].

### 2.12. Statistical Analysis

SPSS 24.0 for Windows (SPSS, Inc., Chicago, IL, USA) was used for data analysis. We analyzed continuous variables and compared them using a t-test, and categorical variables were compared using X^2^ and Fisher exact tests, as applicable. Normally distributed continuous data were expressed as means (SDs) and were assessed using the analysis of variance (ANOVA), independent sample *t*-test, or paired t-test. Nonparametric data were analyzed using the Mann–Whitney and Wilcoxon tests. Two-sided tests were performed to declare statistical significance at *p* < 0.05.

## 3. Results

There were 233 patients initially enrolled in our study, who were patients with histopathological and immunohistochemical breast cancer diagnoses. After completing a selection based on this study’s criteria, we retained a final batch of 137 patients who underwent an SLNB through different methods, including single tracer (colorimetric method, radioisotope), dual tracer, or triple tracer. Taking into account the general features of the batch, we highlighted the fact that 63.5% (*n* = 87) of the cases were under 50 years old, whereas 36.49% (*n* = 50) were above 50 years old, with an average age of 47.3 ± 1.2. Regarding the location of the tumor, in 77.37% of the cases, we identified a unifocal pattern, in 16.05%, we identified a multifocal pattern, and in the rest of the cases, i.e., 6.56%, we identified a multicentric pattern. The histopathologic analysis showed that 88.23% of the cases were characterized as being ductal carcinoma under invasive form or in situ, 11.77% as being lobular carcinoma with a tumoral grading predominantly moderate, distinguished G2 in 51.82% of the cases, 32.84% having a well to weak pattern, and distinguished G3 in 15.32%, according to the Broders classification.

The batch distribution according to the cTMM indicated that 54.01% of the tumors were classified as cT1, 39.41% as cT2, and a notably low 6.56% as cTis. We evaluated breast density for each case individually and standardized the results based on the Tabar-Gram classification, yielding the following findings: 40.87% (*n* = 56 cases) were categorized as level IV—fibro-nodular predominant, 30.65% (*n* = 42 cases) as level III, and 20.43% (*n* = 28 cases) as level V. For cases classified as having fibro-adipose and adipose density, the percentages were below 10%, with level II accounting for 1.45% (*n* = 2 cases) and level I for 6.58% (*n* = 9 cases). (See Table 1).

The immunohistochemical characteristics of the study cohort were classified according to the St Gallen classification. This revealed that 49.63% of the cases were categorized as Luminal A, 28.46% as Luminal B, 16.78% as triple-negative breast cancer (TNBC), and 5.1% as HER2+. Regarding the body mass index (BMI), 44.53% of the patients had a BMI over 30 kg/m^2^, while 55.47% were under that threshold. Sentinel lymph node biopsy was performed using standardized techniques, employing various identification methods (Figure 1). The choice of tracer for identifying the sentinel node was random, resulting in six study subgroups: Group 1 included 24 patients (17.51%) who received indocyanine green (ICG) as the sole tracer; Group 2 comprised 23 patients (16.78%) who were given methylene blue (MB); Group 3 included 18 patients (13.13%) who underwent dual tracer methods using ICG and MB; Group 4 consisted of 21 patients (15.32%) who received Tc-99m and MB; Group 5 had 28 patients (20.43%) treated with Tc-99m and ICG; and Group 6 included 23 patients (16.78%) who were administered a triple-tracer method combining Tc-99m, ICG, and MB. (See Table 2).

Taking into consideration the total number of patients included in the initial study (*n* = 233), only 137 of them actually underwent SLNB, which is a percentage of 58.79%. As a result of the histopathological and immunochemical analysis, a total of 47 cases (34.30%) presented SLN+. Out of those 47 cases, 2.91% presented metastasis-type cluster cells, 8.02% presented micrometastasis, and 23.35% presented macrometastasis. The SNL identification rate for the general batch (*n* = 137) was 91.97% (Table 3).

By analyzing the breast density at the study group level, we identified a statistical correlation between breast density I, II, BMI above 30 kg/m^2^, and age above 50 (*p* < 0.05, 0.101 [0.98, 0.197]; z = 3.82). The influence of a high BMI and age above 50 is explained through the limited lymphatic drainage caused by a growth of the extra visceral adipose tissue, SAT, which comes as a result of the slowdown of the metabolic system and menopause. One study on the distribution and microstructure of lymph vessels was published in 2018 by Patricia de Albuquerue Garcia Redondo, which shows the fact that the growth of subcutaneous adipose tissue minimizes lymphatic drainage [12]. The data presented in Table 4 shows a linear correlation between the identification rate (IR), grade III-IV breast density (based on the Tabar-Gram classification), age under 50, and a BMI under 30 kg/m^2^. The study group’s identification rate was 91.97% (126/137). On the other hand, we did not obtain any significant statistics in regard to the linear correlation between IR and the molecular profile, whether we referred to the luminal subtypes (Luminal A and B), the non-luminal subtypes (HER2+ and TNBC), *p* > 0.05, 0.201 [0.88, 0.167]; z = 1.82, or the tracer type used.

If we look at the identification rate of the sentinel node, in correlation with the breast density, we obtained some significant statistics for the patients that had a positive report between the node structure and the perinodular adipose structure (grade III-IV-V of the Tabar-Gram classification). By analyzing Table 5, we can notice a statistically significant positive correlation of the linear type between the identification rate and the patients with breast density of grades II, IV, and V (0.117 [0.078, 0.228]; z = 4.72; p3 = 0.047, 0.111 [0.98, 0.277]; z = 9.82; p4 = 0.036, and 0.127 [0.088, 0.218]; z = 5.82; p5 = 0.05, respectively). For the patients with adipose breast density, with a subunit report between the breast structure and adipose tissue, we obtained negative statistics between the identification rate and the breast density grade I, II (p1 = 0.03, p2 = 0.02); in this way, we confirmed that the more abundant the quantity of the adipose tissue, the lower the identification rate of the sentinel node. In 2021, Elske Quak et al. published a study in which they highlighted that breast density was found to be significantly correlated with both age (*p* = 0.0004) and BMI (*p* < 0.0001), with lower densities observed in older and heavier patients [13,14]. A thorough analysis of breast density among 821 Finnish women with breast cancer revealed that density categories were linked to both age and BMI [15,16,17,18]. Specifically, older and more obese patients were predominantly found in the lower-density categories (Tabar-Gram I or II). In clinical practice, when dealing with an elderly female patient with larger, fattier breasts for the SLN procedure, it is often accurate to assume that the likelihood of non-visualization is high.

Examining the identification rate (IR) of the sentinel node in relation to the tracer type used, we found an IR of 87.5% when ICG was used as a single tracer (*n* = 24), with an average of 4.02 ± 2.3 nodes excised and a total of 111 nodes, resulting in a sentinel lymph node (SLN) positivity rate of 29.72%. In the group where MB was the sole tracer, the IR was 82.6%, with an average of 2.4 ± 1 nodes excised and a total of 67 nodes, leading to an SLN positivity rate of 19.4%. For the dual tracer technique, the IR was 94.4% for ICG + MB, 90.4% for RI + MB, and 96.4% for RI + ICG. Variations in the average number of excised nodes were also observed: 2.6 ± 1.35 nodes for the ICG + MB group, 3.2 ± 1.56 for the RI + MB group, and 2.34 ± 1.2 for the RI + ICG group (*p* < 0.05). Notably, the triple-tracer method (RI + ICG + MB) achieved an identification rate of 100%, with an average of 3.24 ± 1.54 nodes excised and a total of 108 nodes, resulting in an SLN positivity rate of 24.07%. Statistical analysis revealed significant differences in the total number of excised nodes between the single-tracer (ICG) and triple-tracer (RI + ICG + MB) methods, with Mann–Whitney test results of p1 = 0.002 and p2 = 0.004. However, no statistically significant correlation was found between the positivity rates of SLN and the tracer used for identification (*p* < 0.05) (See Table 6).

The exhaustive analysis of the excision tools (SLNB) emphasized the fact that there were no statistically significant data regarding the average number of nodes in relation to the tracer used (*p* = 0.074), while the incidence of the metastatic measurements of the excised nodes was similar, with an average of 0.263 (X^2^ = 0.38^, *p* = 0.697). We also obtained an average percentage of 87.5% for the number of cases that underwent an excision for more than one sentinel node. This happened with a higher frequency for the groups of patients where we used the RI + ICG tracer (94%), and RI + ICG + MB (97%), obtaining, in this way, a statistically significant correlation *p* = 0.04 (Table 7).

## 4. Discussion

If we refer to the SNL identification technique using the dual tracer, we obtained an IR of 94.4% while using ICG + MB, 90.4% while using RI + MB, and an IR of 96.4% while using RI + ICG. We could show small variations in terms of the average number of excised nodes, obtaining an average number of 2.66 ± 1.35 nodes for the batch where ICG + MB was applied, 3.2 ± 1.56 for the RI + MB batch, and 2.34 ± 1.2 for the RI + ICG batch (*p* < 0.05). For the group where we used the triple-tracer method, that is, RI + ICG + MB, we obtained an IR of 100%, with an average number of nodes of 3.24 ± 1.54, a total number of 108 nodes, and an SNL positivity rate of 24.07%.

An important observational study by Shuo Sun et al. analyzed the detection rate of the sentinel node using tracer methods, including carbon nanoparticles (CNPs), indocyanine green (ICG), and methylene blue (MB), both as a single tracer and in other combinations. Their study included a total of 123 cases of breast neoplasia in the incipient stage cT1-2N0M0, divided into three group studies (SLNB using CNP, CNP + MB, and ICG + MB). The SLNB detection rate for the Shuo et al. study group was similar to ours, with a rate of 97.4% (CNP), 97.6% (CNP + MB), and 95.5% (ICG + MB), without obtaining any statistically significant data (*p* < 0.05) [19,20,21].

A prospective observational study by Wang et al. included 138 patients with breast cancer in an incipient stage who underwent lymphatic mapping and SLNB using ICG and Tc-99mm as tracers. In total, 71 patients were included in a group that underwent SLNB using ICG, while 67 were in a group that underwent SLNB using Tc-99m [22,23,24]. The detection rate was 100% for both groups, with a total of 13 cases that were detected with positive sentinel nodes for the ICG group, while for the Tc-99m group, a total of 12 cases were registered (18.31% vs. 17.91%, *p* = 0.61). Regarding the number of excised nodes, the authors did not obtain any statistically significant data between the two groups (3.12 ± 2.01 vs. 3.33 ± 2.24, *p* = 0.37).

The study developed by Qiu-hui Yang et al. analyzed a total of 300 cases that underwent SLNB using ICG + MB and MB as single tracers. The detection rate of the sentinel node for the two groups was 98.5% (ICG + MB) vs. 91.5% (MB) with *p* = 0.007. The average number of excised nodes for the ICG + MB group was 3.1 vs. 2.6 for the MB group, with *p* = 0.004. Their results indicated that ICG is more accurate in the identification of the axillary lymph nodes [25,26,27,28].

Another meta-analysis published by Martha S. Kedrzycki et al. included 944 patients out of a total of 1748 that were initially considered for the study, with only 10 that actually respected the inclusion criteria of the meta-analysis. The authors concluded that node mapping using ICG is 8.9 times more efficient in SLNB identification than using MB as a single tracer [29,30,31]. Another important conclusion was that they did not obtain any statistically significant data regarding sentinel node identification using ICG compared to the radioisotope technique. As for the number of nodes excised per patient, there were no statistically significant data between the tracers used individually (ICG vs. MB, *p* = 0.051, and ICG vs. Tc-99m, *p* = 0.48). In the study by Goonawardena et al., data from 2301 patients across 19 studies were analyzed. The results showed no significant difference in sentinel lymph node (SLN) detection between indocyanine green (ICG) and radioisotope (RI) methods (odds ratio [OR] 0.90, 95% confidence interval [CI] 0.66–1.24) or in sensitivity (OR 1.23, 95% CI 0.73–2.05), with notable heterogeneity among the studies (I^2^ = 58%, *p* = 0.003). However, dual mapping using both ICG and RI demonstrated significantly improved sensitivity compared with single mapping with RI (OR 3.69, 95% CI 1.79–7.62) or with ICG alone (OR 3.32, 95% CI 1.52–7.24), showing no heterogeneity across studies (I^2^ = 0%, *p* = 0.004).

Sanjit Kumar Agrawal et al. developed a study that included 207 patients with breast cancer (T1-3N0), in which they analyzed the detection rate of SLNB using ICG + MB in conjunction with the standard dual technique, which uses MB+ radioisotope (Tc-99m) as a tracer. The detection rate in their study was 96% (199/207 cases), while if divided, they obtained an identification rate of the sentinel node of 95% for the group with TC-99 + MB and 97% for the group with ICG + MB. The average number of detected nodes was 3.17 ± 1.84 for the TC-99 + MB group, while for the other group, in which the ICG + MB tracer was used, the number was 2.73 ± 1.55. At the same time, for the TC-99 + MB group, 31.3% of the SLNBs were positive, while for the other group, a percentage of 28% were positive [32,33,34,35].

Regarding the identification rate of the sentinel node in correlation with breast density, we obtained statistically significant results for those patients who had a supraunitar report between the glandular and adipose structure (grade III-IV-V of the Tabar-Gram classification). Table 5 highlights a statistically positive correlation of the linear type between the identification rate and the patients with a breast density of grades III, IV, and V. For the patients with an adipose breast density with a subunitar report between the glandular structure and adipose tissue, we obtained a statistically significant negative correlation between the identification rate and breast density grade I, II, thus confirming that the more abundant the quantity of periglandular adipose tissue, the lower the identification rate of the sentinel node.

After performing the histopathologic and immunohistochemical analysis, we identified a total of 47 cases (34.30%) that presented an SLN+. Out of those 47 cases, 2.91% presented metastasis-type cluster cells, 8.02% presented micrometastasis, and 23.35% presented macrometastasis. The SLN identification rate for the general batch was 91.97% (*n* = 137). We registered an average percentage of 87.5% for the cases that underwent an excision for more than one sentinel node. A study published in 2018 by Çolakoğlu MK et al. concluded that the overall success rate for SLN detection was 83.3% for all patients. We detected at least one SLN in 239 of 287 patients and could not find any node in 48 (16.7%) patients [36,37].

We did not obtain any statistically significant data regarding linear-type correlations between the IR and the molecular profile, whether we referred to the luminal subtypes (Lumina A and Luminal B) or to the non-luminal subtypes (HER2+ and TNBC), *p* > 0.05, 0.201 [0.88, 0.167]; z = 1.82. The same study mentioned above, published by Çolakoğlu MK et al. in 2018, concluded that patients with TNBC (triple-negative) or HER2-positive breast cancer had a significantly higher risk of sentinel lymph node (SLN) metastasis compared with those in the luminal group, with this risk being up to six times greater, particularly for TNBC patients. The low SLN detection rates observed in this patient group using methylene blue or radiocolloid alone may be linked to this elevated metastasis risk. Lymphatic tumor emboli could contribute to problems with lymphatic drainage. In contrast, the current study achieved very successful SLN detection rates in the Luminal A/B group, even with the use of the radiocolloid alone.

Sanjit Kumar Agrawal et al. worked on a study that included 207 patients with breast cancer (T1-3N0), in which they performed an analysis of the SLNB detection rate using ICG + MB in comparison to the standardized dual technique that uses MB+ radioisotope (Tc-99m) [38]. The global detection rate in their study was 96% (199/207). When divided, they obtained an identification rate of 95% for the group with Tc-99 + MB and 97% for the group with ICG + MB. The average number of nodes identified was 3.17 ± 1.84 for the first group, while for the second group, which used ICG + MB, the average number was 2.73 ± 1.55. Moreover, for the batch that used Tc-99 + MB as a tracer, 31.3% of SLN were positive, whereas for the batch that used ICG + MB as a tracer, 28% were positive.

There is a big heterogeneity in the specialized literature regarding the gold standard method for sentinel node identification as well as lymphatic mapping. A lot of guidelines recommend the dual technique, which is represented by using Tc-99m and blue dye/methylene blue in the process of identifying the sentinel node. However, some studies emphasize the cardinal role of ICG as a unique or associated method. If we refer to the most important international guidelines, it can be observed that the American guidelines (American Society of Breast Surgeons) do not yet include ICG as a possible tracer choice used in performing lymphatic mapping [39]. On the other hand, the Asian (Chinese Society of Breast Surgery and Japanese Breast Cancer Society) and European (European Society of Medical Oncology—ESMO) guidelines support using ICG for lymphatic mapping as well as for SLNB. The American Society of Breast Surgeons, in their Performance and Practice Guidelines for SLNB, supports the usage of a dual technique based on radioisotope (RI) combined with methylene blue (MB) and claims to have superior results. However, the American Society of Clinical Oncology (ASCO) does not mention using ICG to detect the sentinel node. Contrary to what the American guidelines mention and recommend, the ESMO guidelines recommend the usage of ICG as an alternative to the dual technique using RI + MB, with an SLN identification rate of above 97% and a negative detection rate (FNR). Similar to the European guidelines, the Asian guidelines (Chinese Society of Breast Surgery and Japanese Breast Cancer Society) also recommend using ICG + MB as a standard technique in performing lymphatic mapping and SLNB. The recent findings by Redonda et al. highlight the rarity of lymphatic capillaries in human subcutaneous adipose tissue, which aligns with our observation that breast density significantly impacts the effectiveness of lymphoscintigraphy protocols. In this article, we aimed to emphasize the relationship between breast density and the non-visualization of sentinel lymph nodes (SLNs), complementing the insights provided by Chahid et al. We believe it is an opportune moment to initiate a discussion on the potential role of breast density—readily assessed through mammography and breast MRI, which patients typically undergo during baseline evaluations—in adapting SLN protocols [40].

The present article has some limitations related to the number of patients included in this study. However, it still represents a launching pad for future fundamental research aiming to identify a gold standard procedure for performing SLNB in accordance with certain factors with predictive potential such as glandular density, BMI, age, and other analytical parameters under research.

## 5. Conclusions

Regarding the identification methods for sentinel nodes, there is no gold standard method, but by analyzing the results of our study, we could claim that the triple-tracer method (RI + ICG + MB) offered the most accurate results and permitted the most identification of the sentinel node. Breast density represents a predictive biomarker of the identification rate of the sentinel node, having a direct correlation with BMI (above 30 kg/m^2^) and an age of above 50 years. The classification of the patients according to breast density represents an important criterion, where an adipose breast density (Tabar-Gram I-II) represents a lower IR of SLN compared with a density of the fibro-nodular type (Tabar-Gram III-V). We did not obtain any statistically significant data for the linear correlations between IR and the molecular profile, whether we referred to the luminal subtypes (Luminal A and Luminal B) or if we referred to the non-luminal subtypes (HER2+ si TNBC), *p* > 0.05, 0.201 [0.88, 0.167]; z = 1.82.

## Figures and Tables

**Figure 1 diagnostics-14-02439-f001:**
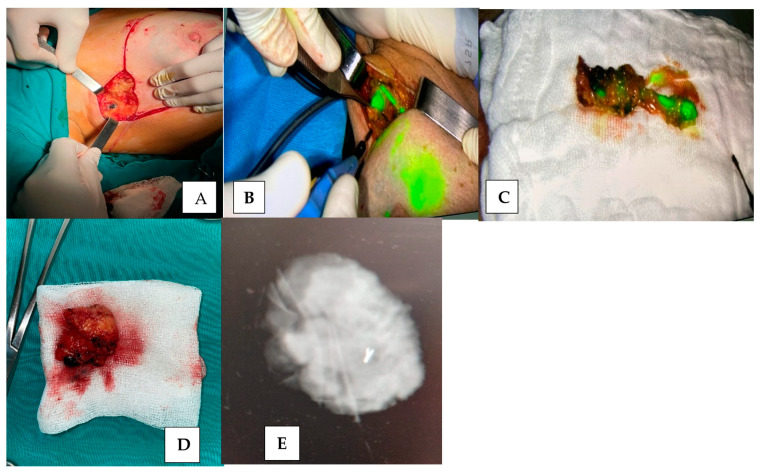
(**A**–**E**) SLNB identification using different methods of identification and marking (personal archive)**.** (**A**)—sentinel lymph node biopsy (SLNB) using MB (personal archive); (**B**)—sentinel lymph node biopsy (SLNB) using ICG (personal archive); (**C**)—SLNB using MB + ICG; (**D**,**E**)—target axillary dissection (TAD)—final aspect of excision piece/mammographic appearance of excision.

**Table 1 diagnostics-14-02439-t001:** General characteristics of the batch.

Age	No. of Patients	%
Under 50	87	63.5
Above 50	50	36.49
**Grading tumoral (Broders classification)**		
G1	45	32.84
G2	71	51.82
G3	21	15.32
**Lympho-vascular invasion**		
Yes	44	32.11
No	93	67.88
**Breast density—Tabar-Gram**		
**I**	9	6.58
**II**	2	1.45
**III**	42	30.65
**IV**	56	40.87
**V**	28	20.43
**Surgery type**		
BCS	72	52.55
OBCS	24	17.5
Mastectomy	41	29.92
**Tumor status**		
Unifocal	106	77.37
Multifocal	22	16.05
Multicentric	9	6.56
**Molecular subtype**	**No. of patients**	**%**
Luminal A	68	49.63
Luminal B	39	28.46
HER2+	7	5.1
TNBC	23	16.78
**cTNM**	**No. of patients**	**%**
Tis	9	6.56
T1	74	54.01
T2	54	39.41

**Table 2 diagnostics-14-02439-t002:** Batch distribution based on the tracer type used for SLN identification.

Variables	No. of Patients	%
**Molecular subtype**		
Luminal A	68	49.63
Luminal B	39	28.46
HER2+	7	5.1
TNBC	23	16.78
**BMI**		
<30 kg/m^2^	76	55.47
>30 kg/m^2^	61	44.53
**Tracer**		
ICG	24	17.51
MB	23	16.78
ICG + MB	18	13.13
RI + MB	21	15.32
RI + ICG	28	20.43
RI + ICG + MB	23	16.78
**TAD (Targeted axillary dissection)**	**36**	**26.27**

**Table 3 diagnostics-14-02439-t003:** Identification rate and SLN histopathological characteristics.

Variables	No. of Patients	%			
**Total number of patients**	233				
**No. of SLNB**	137	58.79%			
**SLN+**	47	34.30%	Cluster cells *n* = 4	Micrometastasis *n* = 11	Macrometastasis *n* = 32
**SLN-**	90	65.70%	2.91%	8.02%	23.35%
**Identification rate (IR)**	126/137	91.97%			

**Table 4 diagnostics-14-02439-t004:** The imagistic–molecular characteristics of the batch according to the SLN identification method.

Parameter	ICG (*n* = 24)	MB (*n* = 23)	ICG + MB (*n* = 18)	RI + MB (*n* = 21)	RI + ICG (*n* = 28)	RI + ICG + MB (*n* = 23)	*p*
Luminal A	12	11	8	12	14	11	0.0678
Luminal B	9	6	6	7	6	5	0.065
HER2+	2	1	0	1	1	2	0.156
TNBC	1	5	4	1	7	5	0.345
**BMI**							
<30 kg/m^2^	17	11	8	13	12	15	0.043
>30 kg/m^2^	7	12	10	8	16	8	0.076
**Age**							
Under 50 years	12	19	14	14	9	19	*p* < 0.05
Above 50 years	12	4	4	7	19	4	
**Breast density Tabar-Gram**							
**I**	2	2	2	1	1	1	*p* = −0.025
**II**	1	0	0	0	0	1	
**III**	2	6	7	4	16	7	*p* < 0.05
**IV**	10	9	6	12	8	11	
**V**	9	6	3	4	3	3	
**IR (identification rate) 126/137**	**IR global = 91.97%**						
Yes	21	19	17	19	27	23	*p* < 0.05
No	3	4	1	2	1	0	

**Table 5 diagnostics-14-02439-t005:** Correlation between breast density (according to the Tabar-Gram classification) and the identification rate of SLN.

Breast Density Tabar-Gram	ICG (*n* = 24)	IR1 21/24	MB (*n* = 23)	IR2 19/23	ICG + MB (*n* = 18)	IR3 17/18	RI + MB (*n* = 21)	IR4 19/21	RI + ICG (*n* = 28)	IR5 27/28	RI + ICG + MB (*n* = 23)	IR6 23/23	*p*
**I**	2	0	2	0	2	1	1	0	1	0	1	1	−0.03
**II**	1	1	0	0	0	0	0	0	0	0	1	1	−0.02
**III**	2	1	6	4	7	7	4	3	16	16	7	7	0.047
**IV**	10	10	9	9	6	6	12	12	8	8	11	11	0.036
**V**	9	9	6	6	3	3	4	4	3	3	3	0.05	0.05

**Table 6 diagnostics-14-02439-t006:** SNL Identification rate depending on the group studies.

Tracer	IR (%)	Average No. of Nodes	Total No. of Nodes	SLN + (SLN Positivity Rate)
**ICG**	87.5	4.02 ± 2.3	111	33/111 (29.72)
**MB**	82.6	2.4 ± 1.1	67	13/67 (19.4)
**ICG + MB**	94.4	2.66 ± 1.35	66	17/66 (25.75)
**RI + MB**	90.4	3.2 ± 1.56	89	26/89 (29.21)
**RI + ICG**	96.4	2.34 ± 1.2	94	31/94 (32.97)
**RI + ICG + MB**	100	3.24 ± 1.54	108	26/108 (24.07)

**Table 7 diagnostics-14-02439-t007:** SLNB performance between study groups.

Parameter	ICG (*n* = 24)	MB (*n* = 23)	ICG + MB (*n* = 18)	RI + MB (*n* = 21)	RI + ICG (*n* = 28)	RI + ICG + MB (*n* = 23)	*X* ^2^	*p*
**SLN IR**	21/24	19/23	17/18	19/21	27/28	23/23		0.042 *
**Mean No. of SLNs (SD)**	4.02 ± 2.3	2.4 ± 1.1	2.66 ± 1.35	3.2 ± 1.56	2.34 ± 1.2	3.24 ± 1.54	−1.80 ^	0.074
**Mean metastatic SLN count**	0.29	0.19	0.25	0.29	0.32	0.24	0.38 ^	0.697
**SLN positivity rate**	33/111 (29.72)	13/67 (19.4)	17/66 (25.75)	26/89 (29.21)	31/94 (32.97)	26/108 (24.07)	0.47	0.789
**No. of patients with >1 SLN**	18/21 (86%)	15/19 (79%)	13/17 (81%)	16/19 (88%)	25/27 (94%)	22/23 (97%)	1.17	0.04

* By Fisher exact test ^ *t*-test.

## Data Availability

All data used in this study can be obtained from the author upon request.

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
