# Peer review of "Sentinel Lymph Node Biopsy in Breast Cancer Using Different Types of Tracers According to Molecular Subtypes and Breast Density—A Randomized Clinical Study"

_diagnostics, 2024, doi:10.3390/diagnostics14212439_

Round 1

Reviewer 1 Report

Comments and Suggestions for Authors

Overall, the manuscript is interesting, even though there are published records related to this area. However, there are a few concerns.

1. English editing is required throughout the manuscript.

2. The authors should provide an ethical statement as to whether the patients gave their informed consent or whether the study received a waiver from the consenting process.

3. The discussion is weak and should be supported by more recent evidences from the literature.

4. Survival data and its detailed analysis can be included which will add value to the manuscript.

Comments on the Quality of English Language

English editing is required throughout the manuscript.

Author Response

Response to Reviewer 1 Comments

  1. English editing is required throughout the manuscript.

I made the changes related to the lingvistic character of the article by correcting some obvious mistakes and rewriting some paragraphs

  1. The authors should provide an ethical statement as to whether the patients gave their informed consent or whether the study received a waiver from the consenting process.

All patients included in the study signed a standardized consent form regarding inclusion in the study

  1. The discussion is weak and should be supported by more recent evidences from the literature.

I reformulated certain paragraphs from the discussion section and at the same time added new studies from the literature for an analytical parallelism.

  1. Survival data and its detailed analysis can be included which will add value to the manuscript.

   The results regarding long-term survival and the disease-free interval are being systematized and the preliminary results will be expressed in the next article

Reviewer 2 Report

Comments and Suggestions for Authors

The study presents some interesting findings, but the discussion section repeats a lot of the information from the results section.  It would be helpful to relocate the detailed statistical data (such as percentages and p-values) to the results section, while maintaining a focus on broader interpretations and implications in the discussion. 

The discussion currently lacks a clear transition between topics, jumping between breast density, molecular subtypes, and tracers without a clear progression.

The study presents findings that are clinically significant, but it does not provide sufficient detail on this aspect. Please expand the discussion of clinical implications. 

While the study compares different tracers (e.g., ICG, MB, Tc-99m, and their combinations), it does not address the practicality, availability, or cost of these tracers.

In some groups, especially those examining tracer combinations, the sample sizes are relatively small, with only 18 patients in the ICG+MB group. Please acknowledge this limitation more explicitly in the discussion section.

The article makes reference to the Tabar-Gram classification of breast density at an early stage, but does not provide a comprehensive explanation of its significance until much later.  To provide a clearer background, please introduce the concept of breast density and its impact on SLN identification earlier in the article.

There are formatting issues, such as repeated phrases and some references missing full citation details, and spelling mistakes in tables. 

Comments on the Quality of English Language

Throughout the article, terms like “corelation” and “corelations” appear, which should be spelled as "correlation" . 

Many sentences in the article are unnecessarily long and complex, making the text difficult to follow.  Simplifying such sentences would improve readability and understanding.

Author Response

Response to Reviewer 2 Comments

Thank you very much for taking the time to review this manuscript.

1.The discussion currently lacks a clear transition between topics, jumping between breast density, molecular subtypes, and tracers without a clear progression.

I made the extension of the discussions in outlining some clear directives, I rewrote the non-conclusive paragraphs

2.The study presents findings that are clinically significant, but it does not provide sufficient detail on this aspect. Please expand the discussion of clinical implications. 

I made the modification

3.While the study compares different tracers (e.g., ICG, MB, Tc-99m, and their combinations), it does not address the practicality, availability, or cost of these tracers.

The analysis of the costs of the methods of performing SLNB was not the object of study of the article, but we presented the different methods of performing SLNB using the single-double or triple tracer method, each with different detection rates, taking into account certain predictive factors such as glandular density

4.In some groups, especially those examining tracer combinations, the sample sizes are relatively small, with only 18 patients in the ICG+MB group. Please acknowledge this limitation more explicitly in the discussion section.

I made the modification

5.The article makes reference to the Tabar-Gram classification of breast density at an early stage, but does not provide a comprehensive explanation of its significance until much later.  To provide a clearer background, please introduce the concept of breast density and its impact on SLN identification earlier in the article.

I have detailed the Tabar Gram classification on page 2 of the article specifying the fact that the poor glandular structure with increased adiposity at the glandular level shows a more limited lymphatic carrier component

6.There are formatting issues, such as repeated phrases and some references missing full citation details, and spelling mistakes in tables. 

I made the changes at the level of tables and references.

Comments on the Quality of English Language

Throughout the article, terms like “corelation” and “corelations” appear, which should be spelled as "correlation" . 

I made the modification.

Many sentences in the article are unnecessarily long and complex, making the text difficult to follow.  Simplifying such sentences would improve readability and understanding.

I have reconfigured the linguistic aspects of the article and the way of reproducing the information.

Round 2

Reviewer 1 Report

Comments and Suggestions for Authors

Discussion section is starting abruptly. It needs to be modified. Also minor english language editing needs to be done.

Comments on the Quality of English Language

Minor english language editing needs to be done.
